# Potential Toxic Effects of Airport Runoff Water Samples on the Environment

Anna Maria Sulej-Suchomska [1,*], Piotr Przybyłowski [1] and Żaneta Polkowska [2,*]

1 Department of Quality Management, Faculty of Management and Quality Sciences, Gdynia Maritime University, 81-225 Gdynia, Poland; p.przybylowski@wznj.umg.edu.pl
2 Department of Analytical Chemistry, Faculty of Chemistry, Gdańsk University of Technology, 80-233 Gdańsk, Poland
* Correspondence: a.sulej-suchomska@wznj.umg.edu.pl (A.M.S.-S.); zanpolko@pg.edu.pl (Ż.P.)

**Abstract:** Despite the positive aspects of the intensive development of aviation, airports are considered large-scale polluters. Pollution caused by runoff water (stormwater) is one of the major problems related to airport operations. The aim of this study was to characterize the potential toxic impact on aquatic life from runoff water discharges from four international airports in Europe. Samples of stormwater were collected at airports with different capacities of passenger movement in four seasons of the year from 2011 to 2013. Within the ecotoxicological analyses, a battery of biotests incorporating organisms of different trophic levels (Microtox® test, Thamnotoxkit F™) were used. A relatively high number of runoff water samples collected at the investigated airports in Europe was recorded as having very high acute hazard (16.8%), acute hazard (27.7%), and slight acute hazard (18.1%) levels. The results of the research indicate that winter and autumn present a greater toxic threat than the rest of the year. The highest number of toxic samples was observed for samples collected in the de-icing area, the runway and the vicinity of airport terminals. The ecotoxicological assessment applied in this research can be used as a tool for assessing the environmental effect of airports.

**Keywords:** airport runoff water; airport stormwater; toxicity; environmental quality management; sustainable development; water quality; *Vibrio fischeri*; *Thamnocephalus platyurus*; airport management; management of wastewater

## 1. Introduction

Worldwide, air traffic has experienced a net increase in recent decades [1,2]. Statistics show that passenger air traffic has increased globally at an average rate of 5.3% per year since 2000 [2]. Moreover, the European Union Aviation Safety Agency estimates that the number of flights will increase by 42% from 2017 to 2040 according to the most likely forecast [3]. Despite the many positive aspects resulting from the extensive development of commercial aviation, airports are a major source of environmental pollution [4–8]. In this regard, a crucial aspect is the contamination caused by airport stormwaters (runoff waters). Airport runoff waters are formed when precipitation or atmospheric deposition washes chemicals used during everyday activities at airports off the airport platform. Such waters enter the soil, surface water, and even groundwater, which can act as a source of drinking water [9–14]. This can cause substantial difficulties, especially if the receiving existing wastewater treatment plant (WWTP) is malfunctioning or there is no WWTP at the airport [15,16].

Airport runoff waters can contain a wide range of hazardous pollutants, such as benzotriazoles (BTs), heavy metals, glycols, detergents, polychlorinated biphenyls (PCBs), polycyclic aromatic hydrocarbons (PAHs), pesticides, formaldehydes, and phenols, at different concentrations [17–20]. The aforementioned contaminants can be discharged into runoff

waters during refuelling, fuel storage, fuel transportation, aircraft repair, aircraft and airport apron cleaning, de/anti-icing, and engine aviation fuel combustion operations [21–25]. These groups of pollutants are characterized by a high toxicity and carcinogenicity [26–30]. This topic is a serious problem for a wide range of stakeholders, especially those residing in communities near airports, whose health issues, property values, and life quality metrics can be affected by such environmental impacts [3,31,32].

In this context, airport runoff waters should be collected, treated, and monitored constantly to avoid adverse effects on the environment and, above all, on humans and animals [5,33,34]. The quality of the aquatic environment is traditionally evaluated via chemical or physical analyses. Although fundamental for quantifying pollutants, such analyses rarely assume their composite toxicity directly [35]. Therefore, it is crucial to assess the toxic effect of polluted airport runoff waters on living organisms. Understanding the level of airport runoff toxicity to aquatic organisms is essential for the effective management of stormwater quality [36–40]. Biotests can fully complement the monitoring system based on chemical determination of the most hazardous pollutants in airport runoff water samples. There have been a very limited number of global-scale research studies using bioassays to assess the toxicity of airport runoff waters on aquatic organisms [41,42]. Based on the published literature, it can be summarized that most of the aforementioned works provide data on the possible toxicological effects from aircraft de-icer and anti-icer solutions to various types of aquatic organisms, namely, *Pimephales promela*, *Daphnia magna*, *Daphnia pulex*, *Ceriodaphnia dubia*, *Photobacterium phosphoreum*, *Lemna gibba*, and *Aliivibrio fischeri*. There are clear gaps in the study results of the acute whole effluent toxicity of runoff water from airports. To date, the field of the conducted research is very limited. The test results included infrequently organized sampling campaigns as well as sampling within only one airport in each case. So far, the available reports do not provide data on the comparative analysis of toxicity assessments of airports located in different geographical regions and that are characterized by different levels of activity. Although our preliminary research provides the results of the ecotoxicological effects of various compounds in complex airport effluents, the information contained therein has been very limited in regard to the number of investigated airports, few sampling campaigns, as well as single sampling sites on the platform airports. This research topic is not yet fully understood and requires further research.

The main aim of the present study was to characterize the potential toxicity of runoff water discharges from four international airports in Europe. Runoff water samples were collected at international airports with large, high, medium, and low capacities of passenger movement in four seasons—autumn, winter, spring and summer—in the period from 2011 to 2013. Within the ecotoxicological analyses, a battery of biotests incorporating organisms of different trophic levels (Microtox® test and Thamnotoxkit F™) were used. To our knowledge, this is the first time that very detailed toxicity investigations of airport runoff waters on such a scale regarding different European airports and a variety of drained areas and their seasonal variations have been published.

## 2. Materials and Methods

### 2.1. Runoff Water Sampling, Collection and Handling

The runoff water samples were collected from the areas of four international airports in Great Britain and Poland, coded as follows: Large Airport (UK), Big Airport (PL), Medium Airport (PL), and Small Airport (PL). Average number of passenger movement at the investigated airports was 45 million passengers per year (Large Airport UK), 18 million passengers per year (Big Airport PL), 9.8 million passengers per year (Medium Airport PL), and 0.4 million passengers per year (Small Airport PL). Samples were collected from December 2011 to January 2013 in four seasons—summer, autumn, winter, and spring. During this period, 121 runoff water samples were collected (51 samples—Small Airport (PL), 32 samples—Medium Airport (PL), 33 samples—Big Airport (PL), 5 samples—Large Airport (UK)).

Stormwater samples were mainly collected manually. During the research period, the amount of precipitation ranged from 2 to 10 mm and the events lasted from 3 to 5 h. The samples were collected from surface depressions near drain inlets and from the airport drainage system. A list of chosen airport runoff water sampling sites was created by the authors of this paper using the available protocols of airport industrial waste management, the authors' prior experience, and by consulting a group of experts consisting of airport transport and engineers working at relevant airports [7,16,31,43,44]. A very important criterion taken into consideration in selecting the sampling sites was the air traffic intensity at the airports. The essential criterion was also the possibility of signing an agreement with the airport and obtaining a permit to collect samples from the airport platform. The permission to collect the runoff water samples from the airport platform during everyday operations of the airport maintenance obliges to quick (maximum five minute) sampling from the single measuring point at the airport. These conditions result from the necessity of keeping the regular cycle of work related to the proper airport functioning and more stringent procedures introduced recently and connected with increased air traffic while maintaining the high service standard and the safety level of air operations. As a result, four international airports with different capacities of passenger movement were selected. The sampling locations at the airports were areas with the highest concentration of technical service operations (the de-icing area, airport terminal, machinery storage area, runway, car park, parking area) where the largest amounts of pollutants enter drainage ditches with runoff and may be released into the environment (Table 1). Additionally, one location at each airport was selected at the periphery of the airport or in the immediate vicinity of the airport for comparison purposes.

**Table 1.** Characteristics of airport runoff water sampling sites.

| Airports/Sampling Site | Large Airport UK | Big Airport PL | Medium Airport PL | Small Airport PL |
|---|---|---|---|---|
| 1 | runway | influent of a river | vicinity of an airport terminal | vicinity of an airport terminal |
| 2 | a river in the vicinity of the airport | effluent of a river | the technical road | de-icing area |
| 3 | de-icing area (2) | municipal water catchment area | de-icing area | machinery stock, parking places |
| 4 | de-icing area (3) | CARGO water catchment area | machinery stock, parking places | runway |
| 5 | de-icing area (4) | airport ramp | the periphery of an airport | parking places |
| 6 | the road near the airport | car park | runway | the periphery of an airport |
| 7 | - | de-icing area | car park | car park |
| 8 | - | airport ramp | - | - |

The stormwater samples were collected in 1000 mL dark glass bottles by using a syringe (100 mL) with Teflon tubing. Prior to use, the syringes and tubing were rinsed with ultrapure water, and then stormwater was sampled.

After the sample collection campaigns, the stormwater samples were transported to the laboratory and stored at 4 °C until further analysis.

### 2.2. Toxicity Testing of Airport Runoff Water

The toxicity assessment of airport runoff water samples was carried out using a battery of biotests. Two species from different trophic levels in the food chain were applied, namely, decomposers (bacterium *Vibrio fischeri* (Microtox® test)), and consumers (crustacean *Thamnocephalus platyurus* (Thamnotoxkit F™ test)). These microbiotests do not require the maintenance of continuous cultures of organisms and are based on immobilized or dormant (cryptobiotic) stages of selected aquatic species set free or hatched as needed [10,45,46].

Parameters such as the pH, turbidity, temperature, and colour can interfere with the test results; therefore, these factors were examined before each test.

Toxicity testing with the Thamnotoxkit F$^{TM}$ test was performed according to the manufacturer's standard procedure. Cysts were incubated at 25 °C for 22 h under continuous illumination. Ten larvae of aquatic crustacean *Thamnocephalus platyurus* were added to each well containing 1 mL of airport stormwater sample. The test was carried out in triplicate for each airport runoff water sample. Multiwell plates were incubated in darkness for 24 h at 25 °C. The test reaction is the mortality of the organism. The Thamnotoxkit F$^{TM}$ test was performed in triplicate for each airport runoff water sample.

In turn, the Microtox$^®$ test was performed in accordance with the procedure PN-EN ISO 11348-3:2008 [47] using the Microtox model 500 instrument (Strategic Diagnostic Inc., Newark, NJ, USA) for freeze-dried bacteria. The pH of the runoff water samples was measured at the beginning of every test to ensure that it ranged between 6 and 8. The sensitivity of bacteria was verified regularly with the reference toxicant $ZnSO_4*7H_2O$. The measured parameter was the bacterial luminescence inhibition (% effect) evaluated after 30 min of incubation. The test results were calculated using the manufacturer's MicrotoxOmni programme. The $EC_{50}$ value was determined for toxic runoff samples. Each test was carried out in triplicate. The repeatability of the results was regularly examined, and the coefficient of variation (CV) fell in the range of 10%.

## 3. Results

### 3.1. Toxicity Tests of Runoff Water Samples

### 3.1.1. Microtox$^®$

Figure 1 presents the toxicity towards *Vibrio fischeri* bacteria in stormwater samples collected from 2011 to 2013 (in different seasons) of four international airports in Great Britain and Poland with large, high, medium, and low capacities of passenger movement. Acute toxicity tests were performed on 121 samples collected from the abovementioned airports at various characteristic places (1–8). Generally, the sampling sites were set as places where the most maintenance work was carried out and where the greatest number of pollutants entered the stormwater. The potential toxicity of the collected airport runoff water samples was described by applying a toxicity classification system proposed by Persoone et al. [48]. Classification was performed on the basis of determining the percentage effect (PE) obtained with each of the bioassays. The response of the organism was classified as toxic when the percentage effect was equal to or higher than 20%. Within the implemented research, the highest number of identified toxic samples (36.4%) was observed for samples collected from Big Airport PL (Figure 1c). The results of the performed studies showed that 11.8% of the tested samples collected at Small Airport PL (Figure 1a) and only 9.4% collected at Medium Airport PL (Figure 1b) were toxic to the bacterium *Vibrio fischeri*. Bacterial luminescence inhibition in the range of 4–24% was observed in samples collected at Large Airport UK during all campaigns (Figure 1d). However, it should be emphasized that for technical reasons, a relatively small number of samples collected at Large Airport UK were tested when we compared it with the number of analysed samples taken from the other investigated airports.

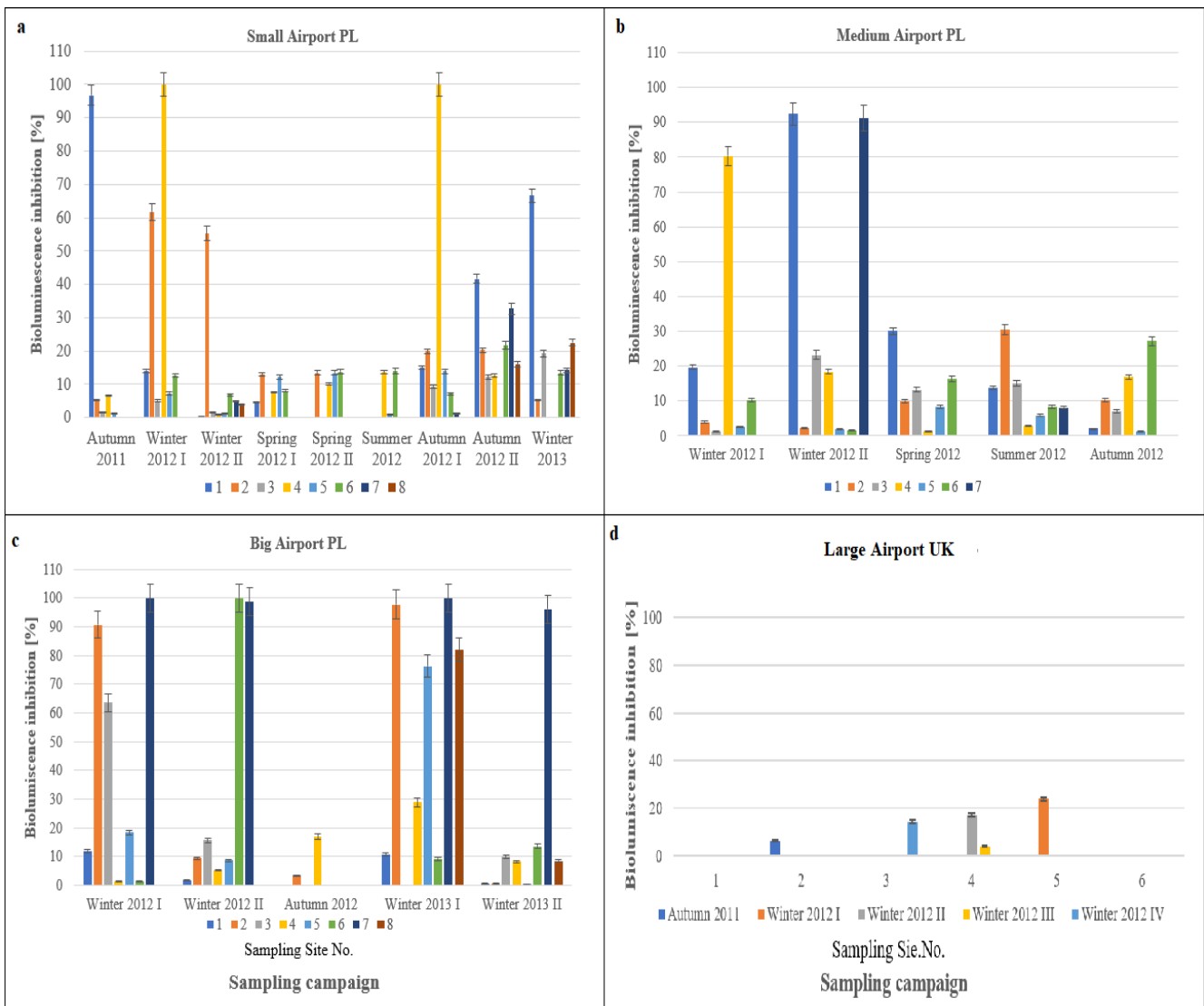

**Figure 1.** Toxicity of airport runoff water samples towards Vibrio fischeri determined by the Microtox® test. (**a**). Small Airport PL; (**b**). Medium Airport PL; (**c**). Big Airport PL; (**d**). Large Airport UK.

On the basis of the obtained data, the highest toxicity towards *Vibrio fischeri* of samples collected at Big Airport PL was determined in samples collected from the effluent of a river from airports, car parks and de-icing areas (91–100% of the bacterial luminescence inhibition) (Figure 1c). In turn, samples collected from the runway, the vicinity of an airport terminal, and the de-icing area (sampling sites no. 4, 1, and 2) at Small Airport PL showed the highest toxicity towards *Vibrio fischeri* (Figure 1a). Similarly, the runoff water samples collected at the Medium Airport PL in the vicinity of the airport terminal, car park and runway were characterized by the highest toxicity (80–92% bacterial luminescence inhibition) (Figure 1b).

Considering seasonal variations in the toxicity of runoff water samples collected at Big Airport PL, Medium Airport PL, and Small Airport PL, it can be concluded that the highest toxicity towards *Vibrio fischeri* was observed in samples collected during winter seasons and autumn seasons, as presented in Figure 1. Based on the results of runoff water toxicity obtained by Microtox®, it could be concluded that the stormwater samples collected in the de-icing area of Big Airport PL during the first winter campaign in 2012 showed the highest toxicity among all tested samples; the $EC_{50}$ values reached 4.96% (Figure 1c). The abovementioned sample (Big Airport PL, winter I, site no. 7) was classified as having

a very high acute hazard level (class V) according to the classification system for acute toxicity proposed by the research team of Persoone et al. [48]; therefore, a dilution of 1:3 of the sample was performed. After sample dilution, the toxicity towards *Vibrio fischeri* was still classified as being of an acute hazard level ($EC_{50}$ = 11.6%). Moreover, the sample (Big Airport PL, 2012, winter I, site no. 7) was ca. 300 times more toxic than the sample collected at the airport ramp (Big Airport PL, 2013, winter II, site no. 5), which was characterized as having the lowest toxicity among the samples tested. Very high toxicity was also observed in the samples collected during the first autumn campaign in 2012 at the runway (Small Airport PL, 2012, autumn I, site no. 4), the winter campaigns at the effluent of a river from the airport (Big Airport PL, 2013, winter I, site no. 2), and the de-icing area of the airport (Big Airport PL, 2013, winter II, site no. 7); the respective $EC_{50}$ values were 6.7%, 7.9%, and 7.43% (undiluted samples). The performed studies showed that 70% of the most toxic samples analysed within this study were collected during the autumn and winter seasons (Figure 1).

### 3.1.2. Thamnotoxkit F$^{TM}$

The results of the Thamnotoxkit F$^{TM}$ test are presented in Figure 2. During all the sampling campaigns, 83 samples were subjected to toxicity assessment towards *Thamnocephalus platyurus*. Toxicity testing with Thamnotoxkit F$^{TM}$ was carried out on runoff water samples collected at the four investigated airports in the 2012–2013 period, taking into consideration the seasonality of sampling campaigns and the characteristic places of airport infrastructure where the most maintenance work was performed. The number of toxic runoff water samples identified by using the Thamnotoxkit F$^{TM}$ test was 93.9% at Big Airport PL, 78.6% at Medium Airport PL, and 96.9% at Small Airport PL among all tested samples (see Figure 2). In the case of runoff water samples collected at Large Airport UK, only one studied sample was classified as toxic when the percentage effect of mortality reached 40% (Figure 2d).

In the present study, the highest toxicity towards *Thamnocephalus platyurus* at Small Airport PL was observed in samples collected in the de-icing area, runway, vicinity of an airport terminal and car park; the respective mortality of crustaceans was 100% (Figure 1a).

When we consider the monitoring of most toxic runoff samples collected at Medium Airport PL using Thamnotoxkit F$^{TM}$, it can be concluded that the highest toxicity was observed (100% mortality of crustaceans) in samples collected in the vicinity of an airport terminal and car park (see Figure 1b). In the present study, a mortality of 100% crustaceans of the species *Thamnocephalus platyurus* was observed in a relatively large number of samples collected at Big Airport PL, inter alia, in the de-icing area, effluent of a river from airports, airport ramps, and car parks (Figure 2c). In turn, the highest toxicity towards *Thamnocephalus platyurus* at Large Airport UK was observed in the samples collected in the de-icing area (Large Airport UK, 2012, winter III, site no. 4), and the crustacean mortality reached 40% (Figure 2d).

Furthermore, Figure 2 presents information on the seasonal variations in toxicity of samples collected at the investigated airports. It is clear from these data that the highest number of toxic runoff water samples (with the highest value of mortality of crustaceans) was observed in samples collected during the autumn and winter seasons. In detail, the mortality in crustaceans of the species *Thamnocephalus platyurus* was in the range of 13.3–60% in samples collected at Small Airport PL during the spring campaign, while samples from the summer, autumn, and winter campaigns were in the ranges of 3.3–60%, 6.5–100%, and 3.3–100%, respectively. In the case of Medium Airport PL, a relatively small number of stormwater samples (i.e., the spring and summer campaigns of 2012) were subjected to toxicity assessment towards *Thamnocephalus platyurus*. Based on the obtained results, it can be concluded that samples collected during both spring and summer campaigns were characterised by a similar toxicity (3.3–100% mortality) towards *T. platyurus* (see Figure 2b). In the case of Big Airport PL and Large Airport UK, due to technical reasons, toxicity testing with the Thamnotoxkit F$^{TM}$ test was performed on

samples collected during autumn and winter campaigns in the period of 2011 to 2013. As mentioned earlier, the obtained results indicate that most of the analysed runoff water samples collected at Big Airport PL were classified as having an acute hazard ($\geq$53.3% mortality of crustaceans) level while having no acute hazard (<20%) level in the case of runoff water samples collected at Large Airport UK, both during the autumn and winter campaigns (Figure 2c,d).

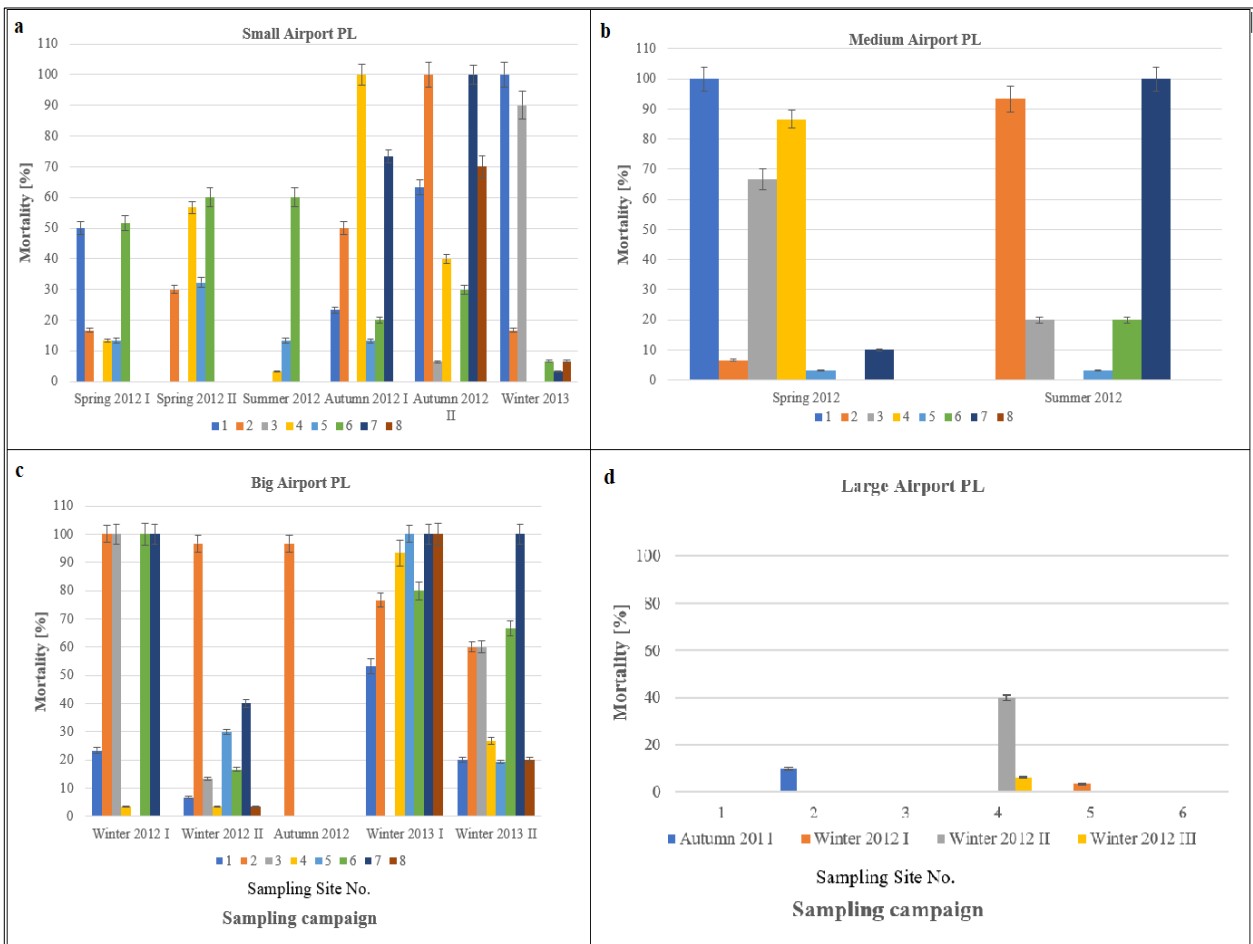

**Figure 2.** Toxicity of airport runoff water samples towards *Thamnocephalus platyurus* determined by the Thamnotoxkit F[TM] test. (**a**). Small Airport PL; (**b**). Medium Airport PL; (**c**). Big Airport PL; (**d**). Large Airport UK.

## 4. Discussion

This research evaluated the potential ecotoxicological effects of airport runoff waters on the bacterium *Vibrio fischeri* and crustacean *Thamnocephalus platyurus* as diagnostic and assessment tools. To our knowledge, bioluminescence bacteria (*Vibrio fischeri*) have been used several times as test organisms in the ecotoxicological testing of runoff water, while the crustacean *Thamnocephalus platyurus* has been used for testing airport runoff toxicity only by the research team of Novak [49] and in our previous work [10]. A decrease in the luminescence of *Vibrio fischeri* or mortality of *Thamnocephalus platyurus* in airport runoff water samples would indicate a potential adverse impact of these environmental samples on aquatic organisms. The results of the measured data presented in Figures 1 and 2 revealed that the crustacean *T. platyurus* and the bacteria *V. fischeri*, two organisms from different trophic levels, respond differently to the target runoff water samples. It can be concluded that the tested organisms are sensitive to different pollutants, which may be present in the investigated airport runoff water samples. This confirms that to assess the real hazard of investigated samples to the environment, the use of various microbiotests is

very helpful and provides a more complete analysis. Due to the importance of this type of ecotoxicological analysis, bioassays have been subjected to detailed investigation and numerous modifications over the last 30 years [48,50,51]. As a result, the measurement procedures have been standardized; thus, there is no need for culturing test organisms, and the developed bioassays are effective and relatively inexpensive. Microtox® is a quick toxicity microbiotest used widely in scientific laboratories as well as in routine analysis. This is mainly due to the simplicity of the operation, the short time of analysis, and there being no need for culturing test organisms. *V. Fischeri* bacteria, which are applied as an indicator organism, are susceptible to wide a spectrum of substances. Nevertheless, the conducted research showed that Microtox® was less sensitive than Thamnotoxkit F™. Many of the investigated airport runoff water samples that were toxic to *Thamnocephalus platyurus* did not show toxic effects towards *Vibrio fischeri*, or the toxic effects were negligible. Similar results in terms of the less toxic response of *Vibrio fischeri* to environmental samples compared to other biotests have been reported in the literature [39,52]. Considering the results of this study and numerous works of other research groups, it should be emphasized that *Thamnocephalus platyurus* is generally more sensitive to surface water and wastewater contamination than *Vibrio fischeri* and other bioindicators applied in bioassays [46,52,53].

The results of the conducted research indicate that winter and autumn present a greater toxic threat than the rest of the year. During the abovementioned seasons, the highest number of operations, such as the de-icing of the airport area and aircraft, is performed, the largest amount of aviation fuel is burned (particularly during take-off when the aircraft engines require more energy and time to start the vehicle due to the reduced ambient temperature), and heat exchanger fluids and chemical stabilizers are more frequently exchanged in comparison with the other warmer seasons of the year, which may account for the above characteristic relationship [10]. Additionally, it can be concluded that samples collected in the de-icing area, car park, vicinity of an airport terminal, and runway were characterized by the highest toxicity, which is also related to the above claim. It was confirmed that places where the most maintenance work at airport platforms is carried out generally presents the greatest toxic threat. Moreover, the results of the bioassays presented herein showed that in some cases, runoff water samples classified as having an acute hazard level were discharged directly to the effluent of a river from airports or penetrated water samples at the periphery of airports. This type of practice at airports poses a very high risk to the environment and to communities near airports.

Based on the results presented in Section 3, an attempt was made to classify the collected airport runoff water samples according to the five-level toxicity classification system proposed by Persoone et al. [48]. Classification was conducted based on determining the percentage effect (PE), estimated during the test performed on an undiluted sample. Table S1 in the Supplementary Materials shows the hazard/toxicity classification system, which is the basis of the performed classification. Data on the allocation of airport runoff water samples to suitable toxicity classes (toxicity classification in relation to both *T. platyurus* and *V. fischeri*) are summarized in Tables 2–5. The recommended management actions for each sampling site, from which runoff water samples were taken for toxicity assessment, are also summarized in Tables 2–5. As a result of the conducted research, *Thamnocephalus platyurus* turned out to be a more sensitive bioindicator to airport runoff water contamination; therefore, the toxicity classification in relation to crustacean *T. platyurus* is discussed in this section.

**Table 2.** Determination of toxicity in the analysed airport water samples collected at Small Airport PL based on the toxicity classification system proposed by Persoone et al. [48].

| Sampling Date/Season | Campaign | Site No. | Microtox® | | | Thamnotoxkit F™ | | |
| --- | --- | --- | --- | --- | --- | --- | --- | --- |
| | | | Threat Degree | Acute Hazard Classes | Recommended Management Actions | Threat Degree | Acute Hazard Classes | Recommended Management Actions |
| 2011 Autumn | I | 1 | Acute hazard | III | IA [a] | NA [d] | NA | NA |
| | I | 2 | No acute hazard | I | FO [b] | NA | NA | NA |
| | I | 3 | No acute hazard | I | FO | NA | NA | NA |
| | I | 4 | No acute hazard | I | FO | NA | NA | NA |
| | I | 5 | No acute hazard | I | FO | NA | NA | NA |
| 2012 Winter | I | 1 | No acute hazard | I | FO | NA | NA | NA |
| | I | 2 | Acute hazard | III | IA | NA | NA | NA |
| | I | 3 | No acute hazard | I | FO | NA | NA | NA |
| | I | 4 | Acute hazard | III | IA | NA | NA | NA |
| | I | 5 | No acute hazard | I | FO | NA | NA | NA |
| | I | 6 | No acute hazard | I | FO | NA | NA | NA |
| 2012 Winter | II | 1 | No acute hazard | I | FO | NA | NA | NA |
| | II | 2 | Acute hazard | III | IA | NA | NA | NA |
| | II | 3 | No acute hazard | I | FO | NA | NA | NA |
| | II | 4 | No acute hazard | I | FO | NA | NA | NA |
| | II | 5 | No acute hazard | I | FO | NA | NA | NA |
| | II | 6 | No acute hazard | I | FO | NA | NA | NA |
| | II | 7 | No acute hazard | I | FO | NA | NA | NA |
| | II | 8 | No acute hazard | I | FO | NA | NA | NA |
| 2012 Spring | I | 1 | No acute hazard | I | FO | Acute hazard | III | IA |
| | I | 2 | No acute hazard | I | FO | No acute hazard | I | FO |
| | I | 4 | No acute hazard | I | FO | No acute hazard | I | FO |
| | I | 5 | No acute hazard | I | FO | No acute hazard | I | FO |
| | I | 6 | No acute hazard | I | FO | Acute hazard | III | IA |
| | II | 2 | No acute hazard | I | FO | Slight acute hazard | II | CA |
| | II | 4 | No acute hazard | I | FO | Acute hazard | III | IA |
| | II | 5 | No acute hazard | I | FO | Slight acute hazard | II | CA |
| | II | 6 | No acute hazard | I | FO | Acute hazard | III | IA |
| | I | 4 | No acute hazard | I | FO | No acute hazard | I | FO |
| | I | 5 | No acute hazard | I | FO | No acute hazard | I | FO |
| | I | 6 | No acute hazard | I | FO | Acute hazard | III | IA |

**Table 2.** *Cont.*

| Sampling Date/Season | Campaign | Site No. | Microtox® | | | Thamnotoxkit F™ | | |
|---|---|---|---|---|---|---|---|---|
| | | | Threat Degree | Acute Hazard Classes | Recommended Management Actions | Threat Degree | Acute Hazard Classes | Recommended Management Actions |
| 2012 Autumn | I | 1 | No acute hazard | I | FO | Slight acute hazard | II | CA |
| | I | 2 | No acute hazard | I | FO | Acute hazard | III | IA |
| | I | 3 | No acute hazard | I | FO | No acute hazard | I | FO |
| | I | 4 | Acute hazard | III | IA | Very high acute hazard | V | IA |
| | I | 5 | No acute hazard | I | FO | No acute hazard | I | FO |
| | I | 6 | No acute hazard | I | FO | Slight acute hazard | II | CA |
| | I | 7 | No acute hazard | I | FO | Acute hazard | III | IA |
| 2012 Autumn | II | 1 | Slight acute hazard | II | CA [c] | Acute hazard | III | IA |
| | II | 2 | Slight acute hazard | II | CA | Very high acute hazard | V | IA |
| | II | 3 | No acute hazard | I | FO | No acute hazard | I | FO |
| | II | 4 | No acute hazard | I | FO | Slight acute hazard | II | CA |
| | II | 6 | Slight acute hazard | II | CA | Slight acute hazard | II | CA |
| | II | 7 | Slight acute hazard | II | CA | Very high acute hazard | V | IA |
| | II | 8 | No acute hazard | I | FO | Acute hazard | III | IA |
| 2013 Winter | I | 1 | Acute hazard | III | IA | Very high acute hazard | V | IA |
| | I | 2 | No acute hazard | I | FO | No acute hazard | I | FO |
| | I | 3 | No acute hazard | I | FO | Acute hazard | III | IA |
| | I | 6 | No acute hazard | I | FO | No acute hazard | I | FO |
| | I | 7 | No acute hazard | I | FO | No acute hazard | I | FO |
| | I | 8 | Slight acute hazard | II | CA | No acute hazard | I | FO |

[a] IA—immediate action required, [b] FO—further observation required, [c] CA—consideration of the need for action required, [d] NA—not analysed.

Table 3. Determination of toxicity in the analysed airport water samples collected at Medium Airport PL based on the toxicity classification system proposed by Persoone et al. [48].

| Sampling Date/Season | Campaign | Site No. | Microtox® | | | Thamnotoxkit F^TM | | |
|---|---|---|---|---|---|---|---|---|
| | | | Threat Degree | Acute Hazard Classes | Recommended Management Actions | Threat Degree | Acute Hazard Classes | Recommended Management Actions |
| 2012 Winter | I | 1 | No acute hazard | I | FO | NA | NA | NA |
| | I | 2 | No acute hazard | I | FO | NA | NA | NA |
| | I | 3 | No acute hazard | I | FO | NA | NA | NA |
| | I | 4 | Acute hazard | III | IA | NA | NA | NA |
| | I | 5 | No acute hazard | I | FO | NA | NA | NA |
| | I | 6 | No acute hazard | I | FO | NA | NA | NA |
| | II | 1 | Acute hazard | III | IA | NA | NA | NA |
| | II | 2 | No acute hazard | I | FO | NA | NA | NA |
| | II | 3 | Slight acute hazard | II | CA | NA | NA | NA |
| | II | 4 | No acute hazard | I | FO | NA | NA | NA |
| | II | 5 | No acute hazard | I | FO | NA | NA | NA |
| | II | 6 | No acute hazard | I | FO | NA | NA | NA |
| | II | 7 | Acute hazard | III | IA | NA | NA | NA |
| 2012 Spring | I | 1 | Slight acute hazard | II | CA | Very high acute hazard | V | IA |
| | I | 2 | No acute hazard | I | FO | No acute hazard | I | FO |
| | I | 3 | No acute hazard | I | FO | Acute hazard | III | IA |
| | I | 4 | No acute hazard | I | FO | Acute hazard | III | IA |
| | I | 5 | No acute hazard | I | FO | No acute hazard | I | FO |
| | I | 6 | No acute hazard | I | FO | No acute hazard | I | FO |
| | I | 7 | NA | NA | NA | No acute hazard | I | FO |
| 2012 Summer | I | 1 | No acute hazard | I | FO | No acute hazard | I | FO |
| | I | 3 | No acute hazard | I | FO | Slight acute hazard | II | CA |
| | I | 4 | No acute hazard | I | FO | No acute hazard | I | FO |
| | I | 5 | No acute hazard | I | FO | No acute hazard | I | FO |
| | I | 2 | No acute hazard | I | FO | NA | NA | NA |
| | I | 4 | No acute hazard | I | FO | NA | NA | NA |
| | I | 5 | No acute hazard | I | FO | NA | NA | NA |
| | I | 6 | Slight acute hazard | II | CA | NA | NA | NA |

**Table 4.** Determination of toxicity in the analysed airport water samples collected at Big Airport PL based on the toxicity classification system proposed by Persoone et al. [48].

| Sampling Date/Season | Campaign | Site No. | Microtox® | | | Thamnotoxkit F™ | | |
|---|---|---|---|---|---|---|---|---|
| | | | Threat Degree | Acute Hazard Classes | Recommended Management Actions | Threat Degree | Acute Hazard Classes | Recommended Management Actions |
| 2012 Winter | I | 1 | No acute hazard | I | FO | Slight acute hazard | II | CA |
| | I | 2 | Acute hazard | III | IA | Very high acute hazard | V | IA |
| | I | 3 | Acute hazard | III | IA | Very high acute hazard | V | IA |
| | I | 4 | No acute hazard | I | FO | No acute hazard | I | FO |
| | I | 5 | No acute hazard | I | FO | No acute hazard | I | FO |
| | I | 6 | No acute hazard | I | FO | Very high acute hazard | V | IA |
| | I | 7 | Acute hazard | III | IA | Very high acute hazard | V | IA |
| | I | 8 | Acute hazard | III | IA | No acute hazard | I | FO |
| 2012 Winter | II | 1 | No acute hazard | I | FO | No acute hazard | I | FO |
| | II | 2 | No acute hazard | I | FO | Acute hazard | III | IA |
| | II | 3 | No acute hazard | I | FO | No acute hazard | I | FO |
| | II | 4 | No acute hazard | I | FO | No acute hazard | I | FO |
| | II | 5 | No acute hazard | I | FO | Slight acute hazard | II | CA |
| | II | 6 | Acute hazard | III | IA | No acute hazard | I | FO |
| | II | 7 | Acute hazard | III | IA | Slight acute hazard | II | CA |
| | II | 8 | No acute hazard | I | FO | No acute hazard | I | FO |
| 2012 Autumn | I | 1 | NA | NA | NA | Acute hazard | III | IA |
| | I | 2 | No acute hazard | I | FO | Acute hazard | III | IA |
| | I | 4 | No acute hazard | I | FO | NA | NA | NA |
| 2013 Winter | I | 1 | No acute hazard | I | FO | Acute hazard | III | IA |
| | I | 2 | Acute hazard | III | IA | Acute hazard | III | IA |
| | I | 4 | Slight acute hazard | II | CA | Acute hazard | III | IA |
| | I | 5 | Acute hazard | III | IA | Very high acute hazard | V | IA |
| | I | 6 | No acute hazard | I | FO | Acute hazard | III | IA |
| | I | 7 | Acute hazard | III | IA | Very high acute hazard | V | IA |
| | I | 8 | Acute hazard | III | IA | Very high acute hazard | V | IA |
| 2013 Winter | II | 1 | No acute hazard | I | FO | Slight acute hazard | II | CA |
| | II | 2 | No acute hazard | I | FO | Acute hazard | III | IA |
| | II | 3 | No acute hazard | I | FO | Acute hazard | III | IA |
| | II | 4 | No acute hazard | I | FO | Slight acute hazard | II | CA |
| | II | 5 | No acute hazard | I | FO | No acute hazard | I | FO |
| | II | 6 | No acute hazard | I | FO | Acute hazard | III | IA |
| | II | 7 | Acute hazard | III | IA | Very high acute hazard | V | IA |
| | II | 8 | No acute hazard | I | FO | Slight acute hazard | II | CA |

**Table 5.** Determination of toxicity in the analysed airport water samples collected at Large Airport UK based on the toxicity classification system proposed by Persoone et al. [48].

| Sampling Date/Season | Campaign | Site No. | Microtox® | | | Thamnotoxkit F™ | | |
| | | | Threat Degree | Acute Hazard Classes | Recommended Management Actions | Threat Degree | Acute Hazard Classes | Recommended Management Actions |
|---|---|---|---|---|---|---|---|---|
| 2011 Autumn | II | 2 | No acute hazard | I | FO | No acute hazard | I | FO |
| 2012 Winter | II | 5 | Slight acute hazard | II | CA | No acute hazard | I | FO |
| | III | 4 | No acute hazard | I | FO | Slight acute hazard | II | CA |
| | IV | 4 | No acute hazard | I | FO | No acute hazard | I | FO |
| | V | 3 | No acute hazard | I | FO | NA | NA | NA |

When we take into consideration the toxicity towards *T. platyurus*, 37.5% of the analysed runoff water samples collected at the Small Airport PL from 2011–2013 were classified as having no acute hazard (class I) (see Table 2). The results of the performed studies showed that 18.6% of all the tested samples collected at Small Airport PL were recorded as having a slight acute hazard. In turn, 31.3% of monitored runoff water samples were classified as having acute hazard levels (class III), where most of them were taken in the autumn and winter seasons. Nearly 13% of the analysed runoff water samples were recorded as having a very high acute hazard level (mainly in the winter period).

Considering the toxicity classification of runoff water samples collected at Medium Airport PL in relation to *T. platyurus*, it can be concluded that 50% of all analysed stormwater samples were classified as having no acute hazard, 14.3% as having a slight acute hazard, 21% as having an acute hazard, and 14.3% as having a very high acute hazard (Table 3).

In the case of hazard classification of runoff water samples collected in the areas of Big Airport PL in the 2012–2013 period, the highest number of samples (30.3%) was classified as having an acute hazard (class III). A very high acute hazard (class V) was noted for a relatively large number of samples, viz., 24.2% of all analysed samples. In contrast, 18.2% of the runoff water samples taken from the abovementioned airport were classified as toxicity class II, and 27.3% were classified as toxicity class I (Table 4).

Taking into account the samples collected in the areas of Large Airport UK during the period of 2011–2012, the runoff water samples were characterized by a lack of hazard to live organisms (toxicity class I). Only one studied sample collected in the areas of Large Airport UK was recorded as a slight acute hazard in relation to *T. platyurus* (see Table 5). The obtained data show that the level of toxicity determined in runoff water samples collected in the area of Large Airport UK was significantly lower compared to that determined in runoff water samples taken from the other investigated airports. This could be related to the weather conditions. During the research period, ambient temperature in the area of Large Airport UK ranged from 6.4 °C to 8.9 °C, while temperature in the areas of other airports ranged from −3.2 °C to 4.0 °C. It can be associated with burning of less fuel, smaller number of operations such as de-icing of aircraft and the airport area required in higher ambient temperatures.

Due to safety regulations and the necessity of maintaining the regular schedule of work related to the proper airport functioning (e.g., refuelling, airport apron cleaning, de/anti-icing), airport runoff water sampling was not possible in some cases. A particularly small number of samples collected at Large Airport UK were tested. This is due to the fact that at this airport, which is characterized by very large air traffic, we often did not obtain permission to collect the runoff water samples from the airport platform.

Considering the aforementioned results presented in Tables 2–5, it can be summarized that a relatively high number of runoff water samples collected at the investigated airports in Europe were recorded as having a very high acute hazard (16.8%), an acute hazard (27.7%), and a slight acute hazard (18.1%). It is essential to undertake immediate airport management action, especially in places at the investigated airport platforms where the most maintenance work was carried out (i.e., de-icing area, runway, vicinity of an airport terminal, and parking places). Studied samples collected at this kind of sampling site were characterized by having the highest level of hazard to live organisms. The results of the conducted research ensure evidence-based information. Taking into account the obtained results, it can be stated that it is crucial to improve the procedures and operations of investigated airport management in the field of the use of de-icing and cleaning agents, the implementation of more effective treatment technologies and technologies for introducing wastewater into sewage networks.

## 5. Conclusions

Airports are a major source of air, water, and soil pollution. Acute and chronic exposure to all compartments of environmental pollution is harmful to human health with established effects, inter alia, coryza and eye irritation effects, cognitive disorders,

cardiovascular diseases, and even death [22,54]. New types of environmental samples, such as those of airport runoff water, which are characterized by a very complex and sometimes variable matrix compositions, can contain a wide variety of toxic contaminants at various concentration levels. Organic pollutants present in runoff water samples can be metabolized or degraded by microorganisms or become diluted to such a scale that they cannot be detected by using available procedures and apparatuses. Furthermore, the chemical, photochemical, and biological transformations of xenobiotics present in runoff water may lead to the formation of toxic substrates from relatively harmless compounds. In view of this, the use of only chemical approaches to environmental contamination assessments around airports is currently considered insufficient [52]. Identifying a wide variety of harmful contaminants at various concentration levels is a demanding task because this kind of study is time-consuming and very expensive. Moreover, it is not possible to investigate all the possible compounds and their interactions in ecosystems. Ecotoxicological analysis enables the assessment of environmental hazards through the analysis of the overall toxicity of samples and integrates the effect of all pollutants, including additive, antagonistic and synergistic effects [52,55,56].

The conducted ecotoxicological assessment of airport runoff water using a variety of bioindicators confirms that it is an objective and useful tool that can be used for runoff water management in airport areas. The results of the performed bioassays complement current knowledge regarding the potential toxic impact of runoff water streams from different European airports from a variety of drained areas and their seasonal variations. To our knowledge, this is the first investigation of airport runoff water performed on such a scale. Furthermore, the results of the performed research permit the following:

- The creation of a solid database that can be helpful in the rapid assessment of ecological risks associated with this type of wastewater stream;
- a better understanding and estimation of the cause-and-effect relationship of the long-term effects of airport pollutants on the environment;
- an implementation of new airport infrastructure management methods (standards and procedures for reducing sources of pollution, recommended remediation techniques, waste recirculation, and the application of environmentally safe de-icing agents).

Taking the above into account, one can confidently say that this study is characterized by major elements of scientific novelty. The obtained research results have a high potential for innovation in regard to the tools used for the assessment of airport impacts on living organisms. Through this kind of ecotoxicological evaluation of airport runoff water, it is possible to reduce the negative impact of airports on the environment and subsequently may lead to an improved quality of life.

**Supplementary Materials:** The following are available online at https://www.mdpi.com/article/10.3390/su13137490/s1, Table S1: Toxicity classification system according to Persoone et al. [48].

**Author Contributions:** A.M.S.-S.—co-authorship of conceptualization of the research study, planned the research, coordinated the project, collected the data, analysed, and interpreted the data, discussed the results, wrote the paper, prepared all tables and figures, performed revisions of the article; Ż.P.—co-authorship of conceptualization of the research study, discussed the results and commented on the manuscript, discussed the manuscript revisions; P.P.—discussed the results and commented on the manuscript. All authors have read and agreed to the published version of the manuscript.

**Funding:** The APC was funded by Gdynia Maritime University, grant number PZ/04/2021.

**Institutional Review Board Statement:** Not applicable.

**Informed Consent Statement:** Not applicable.

**Acknowledgments:** The authors greatly appreciate the anonymous reviewers of this article for their helpful comments and suggestions and the editors for their hard work.

**Conflicts of Interest:** The authors declare no conflict of interest.

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
