# Peer review of "Potential Toxic Effects of Airport Runoff Water Samples on the Environment"

_sustainability, doi:10.3390/su13137490_

Round 1

Reviewer 1 Report

Dear Authors

This is a rather interesting issue that this paper deals with. Serious aspects of ecotoxicological pollution at airports are emerging.
This work should be seen more as a facilitator for the study of the object and the immediate taking of measures to deal with the problems

Author Response

The authors greatly appreciate Reviewer of this article for the comments and suggestions.

Reviewer 2 Report

Dear Authors and Editors,

The paper entitled: Potential toxic effects of airport runoff water samples on the environment is an interesting and quite innovative work worth publishing. However, it needs to address one major question (1) and some minor revision, in particular:

  1. Sampling protocol is not clear: how many samples were taken? How many repetitions? How were the places chosen (according to which criteria?); the detailed discussion of all consequences of limited sampling should be included and conclusions adjusted accordingly
  2. Nearly 20% of references are works of authors; it is not proper; please provide the accurate references
  3. The English check by a native speaker is needed
  4. The discussion and results can be better presented – underlining key points and transferring some material to the supplement

All in all, this work is interesting and to be accepted after addressing those issues.

Yours sincerely,

Reviewer  

Author Response

(The authors gave the same response as above.)

Reviewer 3 Report

The paper deals with the application of two eco-toxicological tests involving Vibrio Fischeri and Thamnocephalus platyurus to several samples of runoff water collected in areas inside and close to four airports.

The authors must clearly state the difference between this work and 1. The previous study from the same authors cited at line 246 (reference n.9); 2. The other studies cited at line 64 concerning the application of bioassays to assess the toxicity of airport runoff waters on aquatic organisms.

Which is the capacity of passenger movement for the four sampling sites?

Line 83 – the procedure of sampling seems a bit approximate, samples should have been collected after the runoff of X mm of rain, that is depending on the intensity of the rain.

Line 131 – on the basis of what do the authors say that the samples were toxic? In which way this information is related with the figures reported in Figure 1?

Line 155 – how can the EC50 be related to the figures reported in Figure 1?

Lines 157-158 and 289 and Table 2 – the toxicity classification system according to Persoone et al must be better explained, the information reported in Table 2 are not easily understandable. Which is the difference between columns IV and V?

Can the information reported in Tables from 3 to 6 be reported in a more compact and sound way?

Why the average situation in the large UK airport was better than in the other three airports?

Bunches of citations (see, for example, line 46, 49, 50, 64) must be avoided. Say why each reference deserves to be cited.

Please avoid repetitions: see, for example, lines 36-38 AND 46-48; lines 76-80.

Author Response

Dear Professor,

Please find enclosed the typescript of our work entitled “Potential toxic effects of airport runoff water samples on the environment”. We have taken into consideration all remarks and suggestions. All answers and explanations are also attached to the manuscript. The article has been professionally edited for English language by native English speaking editors (Certificate of Springer Nature, No. 62A9-C244-85F9-0118-E86A)

Comments and Suggestions for Authors

The paper deals with the application of two eco-toxicological tests involving Vibrio Fischeri and Thamnocephalus platyurus to several samples of runoff water collected in areas inside and close to four airports.

The authors must clearly state the difference between this work and 1. The previous study from the same authors cited at line 246 (reference n.9); 2. The other studies cited at line 64 concerning the application of bioassays to assess the toxicity of airport runoff waters on aquatic organisms.

After a detailed analysis of the literature, it can be summarized that most of the aforementioned works provide data on the possible toxicological effects from aircraft de-icer and anti-icer solutions to various types of aquatic organisms, namely, Pimephales promela, Daphnia magna, Daphnia pulex, Ceriodaphnia dubia, Photobacterium phosphoreum, Lemna gibba, Aliivibrio fischeri. There are clear gaps in the results of the acute whole effluent toxicity of runoff water from airports. To date, the field of the conducted research was really limited. The test results included infrequently organized sampling campaigns as well as sampling within only one airport in each case. So far, available reports do not provide data on the comparative analysis of airports located in different geographical regions and characterized by different levels of activity. Although our preliminary research provides the results of the ecotoxicological effects of various compounds in complex airport effluents, the information contained therein has been very limited in regard to the number of investigated airports, few sampling campaigns as well as singe sampling sites on the platform airports. This research topic is not yet fully understood and requires further research.

According to the reviewer’s suggestion, issue regarding the difference between this study and other studies has been added.

Which is the capacity of passenger movement for the four sampling sites?

Average number of passenger movement at monitored airports was 45 million passengers per year (Large Airport UK), 18 million passengers per year (Big Airport PL), 9.8 million passengers per year (Medium Airport PL) and 0.4 million passengers per year (Small Airport PL).

According to the reviewer’s suggestion, this data has been added in manuscript.

Line 83 – the procedure of sampling seems a bit approximate, samples should have been collected after the runoff of X mm of rain, that is depending on the intensity of the rain.

According to the reviewer’s suggestion, details about the sampling protocol have been added.

Line 131 – on the basis of what do the authors say that the samples were toxic? In which way this information is related with the figures reported in Figure 1?

The potential toxicity of the collected airport runoff water samples was described by applying a toxicity classification system proposed by Persoone et al. 1. Classification was performed based on determining the percentage effect (PE) obtained with each of the bioassays. The response of the organism was classified as toxic when the percentage effect was equal to or higher than 20%.

According to the reviewer’s suggestion, this aspect has been clarified in the Result section.

(1) Persoone, G.; Marsalek, Blahoslav Blinova, I.; Törökne, A.; Zarina, Dzidra Manusadzianas, L.; Nalecz‐Jawecki, G.; Tofan, L.; Stepanova, N.; Tothova, L.; Kola, B. A Practical and User‐friendly Toxicity Classification System with Microbiotests for Natural Waters and Wastewaters. Environ. Toxicol. 2003, 18 (6).

Line 155 – how can the EC50 be related to the figures reported in Figure 1?

Toxicity was expressed both as percentage of effect (PE) as shown in the Figure 1 and EC50 if calculable (for toxic samples). Additionally, the toxicity values (EC50) obtained for tested samples were discussed in the manuscript.  A similar way of presenting results is also practiced by other research groups  2,3.

(2)              Hurel, C.; Taneez, M.; Volpi Ghirardini, A.; Libralato, G. Effects of Mineral Amendments on Trace Elements Leaching from Pre-Treated Marine Sediment after Simulated Rainfall Events. Environ. Pollut. 2017, 220 (March 2018), 364–374. https://doi.org/10.1016/j.envpol.2016.09.072.

(3)              Melnyk, A.; Kuklińska, K.; Wolska, L.; Namieśnik, J. Chemical Pollution and Toxicity of Water Samples from Stream Receiving Leachate from Controlled Municipal Solid Waste (MSW) Landfill. Environ. Res. 2014, 135, 253–261. https://doi.org/10.1016/j.envres.2014.09.010.

Lines 157-158 and 289 and Table 2 – the toxicity classification system according to Persoone et al must be better explained, the information reported in Table 2 are not easily understandable. Which is the difference between columns IV and V?

According to the toxicity classification system of Persoone et al 1 toxicity class IV (high acute hazard) the PE100 is reached in at last one test, while class V (very high acute hazard), the PE100 is reached in all the test.

[1] Persoone, G.; Marsalek, Blahoslav Blinova, I.; Törökne, A.; Zarina, Dzidra Manusadzianas, L.; Nalecz‐Jawecki, G.; Tofan, L.; Stepanova, N.; Tothova, L.; Kola, B. A Practical and User‐friendly Toxicity Classification System with Microbiotests for Natural Waters and Wastewaters. Environ. Toxicol. 2003, 18 (6).

According to the reviewer’s suggestion, Table 2 has been corrected.

Can the information reported in Tables from 3 to 6 be reported in a more compact and sound way?

The authors considered various possibilities of presenting the information contained in Tables 3-6. However, the current form of these tables seems to be the most readable. Therefore, we decided to keep abovementioned tables in unchanged version.

Why the average situation in the large UK airport was better than in the other three airports?

The obtained data show that the level of  toxicity determined in runoff water samples collected in the area of Large Airport UK was significantly lower compared to that determined in runoff water samples taken from the other investigated airports. It could be related to the weather conditions. During the research period, ambient temperature in the area of Large Airport UK ranged from 6.4˚C to 8.9˚C, while temperature in the areas of other airports ranged from
-3.2 ˚C to 4.0˚C. It can be associated with burning of less fuel, smaller number of operations such as de-icing of aircraft and the airport area required in higher ambient temperatures.

Bunches of citations (see, for example, line 46, 49, 50, 64) must be avoided. Say why each reference deserves to be cited.

According to the reviewer’s suggestion, the detailed literature review has been made once again. As a result, the most important literature items were inserted.

Please avoid repetitions: see, for example, lines 36-38 AND 46-48; lines 76-80.

According to the reviewer’s suggestion, the repetitions in manuscript has been corrected.

The authors greatly appreciate Reviewer of this article for the comments and suggestions.

Round 2

Reviewer 3 Report

The authors addressed in a satisfactory way all the reviewer's comments.

However, I think that the visual quality of tables and figures must be improved before publication.